# Sickle Cell Disease Newborn Screening—An Audit of a Twin Island State Pilot Program

**DOI:** 10.3390/ijns9010014

**Published:** 2023-03-01

**Authors:** Shivon Belle Jarvis, Edda Hadeed, Ketty Lee, Marie-Dominique Hardy-Dessources, Jennifer M. Knight-Madden, Claudine Richardson

**Affiliations:** 1Paediatric Department, Sir Lester Bird Medical Centre, Michael’s Mount, St. John’s, Antigua and Barbuda; 2Gambles Medical Centre, Friars Hill Road, St. John’s, Antigua and Barbuda; 3Laboratory of Molecular Genetics and Inherited Disorders of Red Blood Cell, University Hospital of Guadeloupe, Guadeloupe, F.W.I, France; 4Université des Antilles, UFR Médecine/Campus de Fouillolle-Université Paris Cité, Inserm, BIGR, 97159 Pointe-à-Pitre, France; 5Caribbean Institute for Health Research-Sickle Cell Unit, The University of the West Indies, Mona, Kingston 7, Jamaica

**Keywords:** sickle cell disease, newborn screening, audit, Antigua and Barbuda

## Abstract

The prevalence of Sickle Cell Disease (SCD) within the Caribbean region remains second only to that of West Africa. The Newborn Screening (NBS) Program in Antigua and Barbuda remains heavily dependent on grants, therefore ultimately facing sustainability challenges. Early intervention and implementation of preventative measures post-NBS result in significant improvements in morbidity, quality of life, and survival. This audit reviewed the pilot SCD NBS Program in Antigua and Barbuda from September 2020 to December 2021. A conclusive result was received by 99% of babies eligible for screening, 84.3% of which were HbFA, whilst 9.6% and 4.6% were HbFAS and HbFAC, respectively. This was comparable to other Caribbean countries. Sickle Cell Disease was noted in 0.5% of babies screened, which translates to 1 in 222 live births. Eighty-two percent of mothers were aware of their sickle cell status, compared to 3% of fathers. The importance of instituting a quality improvement team post the initiation of a screening program and the need for a robust public education program have been demonstrated by this audit.

## 1. Introduction

It is predicted, through the combination of population estimates and projections, that by 2050, approximately 400,000 newborns annually will be affected by Sickle Cell Anaemia (SCA) globally [1]. Data suggest that the prevalence of SCD in the Caribbean is second only to Sub-Saharan Africa [2], with the “quality of life and life expectancy varying depending on where children are born and where they live” [1].

SCD, an inherited haemoglobinopathy with an autosomal recessive pattern of inheritance, has as its hallmark chronic haemolysis with concomitant vaso-occlusion [3]. This results in multi-organ injury, an increased risk of severe infections, the potential need for chronic transfusions, and a decreased lifespan. Complications are not limited to the physical but can negatively impact the psychological and social spheres [4]. They also have an economic impact on the family and society.

Haemoglobinopathies form one of the core categories of disorders in the Recommended Uniform Screening Panel as recommended by the Advisory Committee on Heritable Disorders in Newborns and Children [5]. The Pan European Consensus Conference in 2019 concurred that “early diagnosis by NBS together with anti-pneumococcal penicillin prophylaxis and vaccination, coordinated follow up and parental education“ result in reduced morbidity and mortality in childhood [6]. SCD NBS in Europe began in the 1970s in small pockets and became more pervasive over 30 years, initially beginning in England [7]. Universal newborn screening in the USA and Canada has enabled the detection of the most common forms of SCD, reduced mortality by 50% in affected children ages 1 to 4, increased overall life expectancy, and afforded an opportunity to offer genetic counselling with options for future pregnancies. What once began in the 1960s due to unwavering advocacy has now become universal [8].

The Caribbean has embarked on progressive newborn screening, which prior to 2006 was limited to Jamaica and French territories and since then has included Tobago, Grenada, and St. Lucia [9]. Antigua and Barbuda, a twin island state in the Caribbean, with a population of 97,928 [10], through the help of the Caribbean Network of Researchers on Sickle cell Disease and Thalassemia (CAREST), and American University Antigua (AUA), embarked on a pilot program of universal NBS in September 2020, which was free of cost to parents. The prevalence of SCD was unknown in Antigua and Barbuda. It has one primary hospital—the Sir Lester Bird Medical Centre (SLBMC)—offering both routine and critical newborn care with just over 1000 deliveries annually. Prenatal diagnosis via chorionic villus sampling or amniocentesis was not available in the hospital setting. These services were also unavailable for prenatal diagnosis of SCD in the sole private facility on the island, through which less than 10% of the deliveries occur. Screening locally was primarily performed in children considered to be at risk, i.e., with a known parental/family history of SCD. Although it is recognized that SCD follows a Mendelian inheritance pattern and is autosomal recessive [11], it is often difficult to determine the chance of the offspring being affected. Maternal status was often known through antenatal testing; however, anecdotally, paternal status was often not routinely known. This sadly resulted in many children being diagnosed either on further testing after having presented with anaemia or other crises in keeping with haemolysis or vaso-occlusion, which unfortunately did not provide the key opportunity needed for early intervention and prevention. This audit sought to establish quality gaps that may exist in sampling, parental knowledge of their sickle status, and the time frame for parental updates regarding their newborn’s status post receipt of results by the health care provider.

## 2. Materials and Methods

### 2.1. Newborn Screening Process

In September 2020, the Ministry of Health, Wellness and the Environment (MOHWE) signed a memorandum of understanding (MOU) with Le Centre Hospitalier Universitaire de Guadeloupe and CAREST in order to implement a 2-year pilot neonatal screening program in Antigua and Barbuda with the aim of ascertaining the prevalence of SCD in Antigua and Barbuda and the need for having a national newborn screening program. This agreement greatly subsidized the costs supported by the European Regional Development Fund. A policy was created by the clinical staff of the Sir Lester Bird Medical Centre (SLBMC).

Sampling was conducted immediately after birth, with cord blood sampled by institutional policy and applied to Guthrie Cards (GC) provided through CAREST. These GC were labelled by the health care provider with the date of birth, date of sample collection, name of the newborn (I/O mother’s name), gender, gestational age, birthweight, demographics of the mother, and sickle cell genotype of each parent, if known. The blotters were allowed to air dry for 4 h and then stored in a refrigerator at a temperature of 5 ± 3 °C. Data written on blotters were entered into a database daily by a Paediatrician. This information was accompanied by a patient’s unique identification number and additional demographics of the parents, which included address, date of birth, and contact number. The samples from the first to last day of the month of active data collection were then reviewed by a second senior Paediatric physician to ensure that all data fields were filled and that the numbers documented were accurate. Once the quality checks were completed, the samples were packaged and sent via FedEx to Guadeloupe on the first day of the following month. The samples underwent a primary screening test using High-Performance Liquid Chromatography (HPLC) and a second confirmatory step based on agarose gel isoelectrofocusing was performed if abnormal results via the HPLC method were attained. The timeframe before results not in keeping with SCD were received by SLBMC from Guadeloupe was one month via email, whereas results suggestive of SCD were communicated via email 20 days after receipt of blotters, as agreed upon in the MOU. As there were no other standards set within the MOU, the acceptable and achievable standards from the National Health Service (NHS) sickle cell and thalassaemia screening program were utilized as a yardstick for the local screening program [12]. Each standard outlined in the aforementioned guide was assigned a number, had an outlined objective, and detailed the criteria for an achievable versus acceptable standard. If a standard was not specifically stated, then the time frame of a month was utilized.

Once results were received, these were entered into the database by a Paediatrician, with parents/guardians subsequently being called and advised of the results received and how and when to collect the result card. Once a parent was successfully contacted, the date of contact was entered into the database. Prior to COVID-19, parents/guardians collected results in person at the ward/outpatient clinic at SLBMC. Open appointments during working hours were given to parents of newborns with a negative screening result, i.e., HbFA. Once the parent/guardian collected the results, the date of collection was entered into the database. Parents of newborns with positive screens, however, were given a specific date and time to come in for counselling by the Paediatric Consultant. This appointment was communicated via phone. A positive screen for the purposes of this audit fell into one of two categories: Category 1—SCD, which included the following results: HbFS and HbFSC; whilst Category 2 results, which included HbFC and carrier states such as HbFAS and HbFAC. Counselling would include receipt of an educational pamphlet with subsequent arrangements being made for follow-up. If a sample was contaminated, the parent/guardian was advised of the same and asked to come in for the sample to be repeated. Subsequent to the increasing and fluctuating number of COVID-19 cases, however, negative screening results were sent to the community clinics for collection at wellness visits, which are attended by all children in the country at pre-designated ages. Only the date of contacting the parent/guardian was entered into the database. Counselling sessions, however, continued as previously described for clients whose newborns had positive screening results. The policy did not outline the timeframe by which the parents should be contacted with Category 1, Category 2, negative, or other results.

### 2.2. Data Collection

Collection of data took place from 1 September 2020 to 31 December 2021, through the SCD database of the SLBMC. Data were then entered into a data extraction sheet through Excel.

Number of live births was collected from the logbook of births and deaths assigned to the maternity ward. Although there is a national data collection system, there can be a lag in the translation of this statistic to the Health Information Division. Likewise, the information communicated to this division is taken from the aforementioned logbook. Home deliveries upon presenting to SLBMC would be included, likewise deliveries at the sole private facility in Antigua that, by personal communication, has less than 10 deliveries per year.

### 2.3. Statistical Analysis

Data are presented as frequencies, percentages, medians, and means. Chi-squared test was used.

## 3. Results

### 3.1. Coverage

During the time period under review, 1560 babies were eligible for SCD NBS; however, only 1558 were screened, 51% of who were male and 49% female. Figure 1 below outlines further participant details.

Additionally, 10 newborns from the eligible pool for screening at SLBMC were missed during the time period of review, i.e., 10 live births were not screened. The reason for these babies being missed was not documented. Possible reasons may include the baby requiring resuscitation and stabilization post-delivery with the omission of taking a sample once stable, a shortage of staff with competing priorities at the time of delivery resulting in screening being deemed less urgent, or the unavailability of cards at the time of delivery. The data for these babies were not entered into the database but were recognized during quality assurance reviews of the maternity logbook of births. Eight babies (0.5% of all screened newborns) were retested either because of an inconclusive result or suspected contamination with maternal blood. An inconclusive result may occur due to the following reasons: a damaged sample, a low haemoglobin level, insufficient blood collection volume, or haemoglobin A and F levels inconsistent with gestational age.

Standards established by this audit were compared to the acceptable and achievable standards established by the NHS. Coverage of SCD NBS was deemed to be of an acceptable standard if 95% of eligible babies received a conclusive blood spot screening test, whilst an achievable standard accounted for 99% of eligible babies receiving a conclusive test. This audit revealed that 99% of babies screened in Antigua and Barbuda received a conclusive screening test, which was in keeping with what was deemed to be an achievable standard.

### 3.2. Phenotypes

Table 1 below demonstrates the results of the NBS based on Phenotype. The Birth Incidence of SCD for this study period was 4.5 per 1000, i.e., an incidence of 1 in 222 live births. A positive screen can either be Category 1 or 2, as previously outlined. Additionally, 9.6% of newborns screened were found to have FAS (sickle cell trait), whilst 4.6% had FAC (Hb C trait). The X in FAX represents an abnormal haemoglobin different from HbC and HbS and therefore warrants precise identification of that abnormal Hb utilizing another technique, including Molecular Biology. Based on the results displayed in Table 1 below, 11 samples would have met the requirement for repeat testing due to the result either being inconclusive or contaminated; however, only 73% of these newborns had repeat sampling performed. Failure for these samples to be repeated likely included failure to contact parents, failure for the parent to keep an appointment given, or a decision by the parent to have the sample repeated privately.

### 3.3. Timeliness

The timeliness of reporting results from the lab in Guadeloupe to SLBMC and the timeliness of parental updates by SLBMC staff were reviewed. Table 2 outlines standards set through the MOU between SLBMC and the laboratory in Guadeloupe. Standards for the timeliness of parental updates were taken from the aforementioned NHS guide. Reporting of Category 1 results to SLBMC was noted to be below the acceptable standard, where 80% of results were reported to SLBMC within 20 days. It should be noted that the minimum number of days for SLBMC to receive a Category 1 result from Guadeloupe was 14, with a maximum of 27 days and an average of 19 days. Communication of category 2 and negative screening results to parents/guardians was below the acceptable standard, as only 54% of them were updated with these results in less than a month by SLBMC. Additionally, 13% of parents/guardians were not contacted, 33% were contacted after a month of SLBMC receiving results, and the status of communication with 0.3% of parents/guardians was not documented and therefore unknown.

### 3.4. Parental Knowledge

Mothers (M = 79.3, SD = 210) were more likely to know their sickle cell status when compared to fathers (M = 2.8, SD = 3.8); t (1312) = 2.44, *p* = 0.01. Figure 2 below demonstrates that 82% of mothers were aware of their sickle cell status, compared to 3% of fathers. Hb Electrophoresis serves as one of the pre-requisite screening tests in pregnancy. This result is documented in the maternal record and antenatal card, which are reviewed at each visit and once admission is required. The breakdown of parental genotype is reflected in Table 3 below. Parental knowledge of their sickle cell status was unlikely to change by the time of repeat testing; therefore, the knowledge of 1550 parents was analyzed (which excluded the 8 that were retested). Paternal genotype was based on self-reporting.

The significance of a father knowing his sickle cell status based on the mother being a carrier or having SCD is demonstrated in Table 4 below.

## 4. Discussion

Antigua and Barbuda had excellent SCD newborn screening coverage for eligible babies born from September 2020 to December 2021. Screening 99% of eligible babies is commendable and comparative to coverage rates of other countries within the region: Guadeloupe > 98%, Martinique > 99%, Jamaica > 98%, and better than rates in Tobago 96%, Grenada 79%, and St. Lucia 45% [9]. This achievement was also significant given the varying challenges that plagued healthcare systems during the COVID-19 pandemic. It is important to review processes through the use of continuous staff education sessions, as two babies that were screened did not meet the established criteria because they were fresh stillbirths, and ten eligible babies were missed.

Reporting of negative and or category 2 results from the Laboratory in Guadeloupe to SLBMC, and the communication of category 1 results to parents by SLBMC staff subsequent to the institution’s receipt of results, based on established standards, were notably two areas of strength with a standard of 100% being achieved, respectively. A critical review of the processes and limiting factors that impact the processing and communication of category 1 results from the level of the external laboratory to SLBMC would be useful, as only 80% of positive results being communicated to SLBMC were within the standard established within the MOU, that of 20 days from the receipt of samples to the communication of results. The reason for the delayed communication of Category 1 results to SLBMC is unknown. It would be interesting to note the impact of the COVID-19 pandemic on processes, as this might have resulted in competing priorities, thus negatively impacting the process.

A more efficient mechanism of communicating negative and/or category 2 results to parents must be established, as only 54% of parents were updated within one month of the institution receiving results. Anecdotal challenges included the inability to contact parents/guardians, either because of the phone not being answered despite multiple attempts of calling, the number listed within the medical record no longer being in service, or being assigned to a different provider. Although these challenges also impacted the communication of Category 1 results, it is possible that staff made a greater effort to contact these parents/guardians by making more attempts given the implication of the result. A call regarding a Category 2 result was also further removed from the baby’s birth, the impact of which is unknown. Alternate mechanisms would facilitate a more seamless way of updating parents/guardians. This may include relaying results via email or sharing results with primary health care providers, who would then relay information at a wellness visit/childhood immunization appointment. Verifying the validity of the telephone number on record and documenting the community clinic/primary care physician to whom the child will receive follow-up prior to maternal discharge would aid the efficiency of the process. The use of clerical staff to update the database and relay negative results would improve efficiency; with carrier states and positive results still being relayed by clinical staff, as the latter would warrant counselling.

The newborn frequency of sickle cell carriers locally, which is 9.6%, is similar to the following countries: Jamaica 9.74–9.94% (1995–2006 & 2016–2017, respectively), Grenada 9.56%, and Tobago 9.32% [9]. The frequency of HbAC, however, at 4.6%, surpassed that of other countries within the region, including Jamaica, Guadeloupe, Tobago, Grenada, and St. Lucia, with rates ranging from 2.08% to 3.86%. The cumulative carrier state within our twin island state, therefore, was 14.2% (HbAS and HbAC). With the birth incidence of FS and FSC being 0.3% and 0.2%, respectively, this corresponds to 1 in 222 live births having a major SCD syndrome, thus being similar to French Guiana and slightly lower than Jamaica and Tobago [9]. The institution of early childhood care programs for these children, therefore, must be prioritized, given the established benefits previously explored. The impact of foreign mother births was not explored in this audit. Madrid in their review of the first 15 years of their NBS program considered the increase of their rates of Sickle cell trait to be possibly attributed to an increased increment of foreign mother births over the previous two decades [13].

With only 73% of the babies who required retesting being conducted, it would have been an added benefit to capture the reasons for the failure of the retest so that a more comprehensive review could have been performed. This would help to explore whether the limitation was at the level of the client or of the provider.

In addition, 82.5% of mothers know their sickle cell status at the time of delivery, compared to 2.9% of fathers. Of the fathers who knew their sickle cell status, they were more likely to know their status if the mother had sickle cell disease or was a carrier. This speaks to the need for educational campaigns, which will ultimately lead to attaining one of the goals of “Healthy People 2030”, that of achieving health literacy to improve the health and well-being of all [14]. A more robust campaign may therefore include the use of social media, media, outreach, and seminars [15]. Pre-conceptional counselling and testing are key.

There is currently no legislative mandate regarding SCD NBS in Antigua and Barbuda, and in the absence of champions in policy-making positions and continued sponsorship, the sustainability of this program remains grim. Policymakers need to be privy to preliminary reviews of outcomes and see the need for SCD NBS to be a public health measure, within which the importance of educational programs and preventative measures should not be underestimated. The role of the Antigua and Barbuda Sickle Cell Association, a local non-governmental institution, needs to be strengthened. This entity, if appropriately subsidized, could be utilized to decrease the burden on clinicians regarding the processing of samples, entry, and delivery of results to clients, as well as linkage to care post-screening. Linkage to care can be a screening program’s greatest challenge [11]; therefore, it must be ensured that any barrier that exists to preclude this crucial link is explored and appropriately managed.

Quality assurance must include the creation and use of a confirmatory testing protocol to verify the screening result. This does not exist locally but currently exists in the USA and Canada, though policies vary by state [8].

### Limitations

⮚The database failed to capture the timeliness of follow-up for clients who had Category 1 results, including the timeliness of the institution of preventative care. This information, therefore, could not be reviewed.⮚Impact of COVID-19 on the NBS program and available resources were not explored.⮚The incidence of SCD variants captured by this newborn screening program excluded babies born abroad. This would need to be considered for incidence rates on a national level.

## 5. Conclusions

This audit has revealed promising outcomes. Coverage of eligible babies was commendable and comparable to other countries within the region. Although the reporting of negative and category 2 results by the processing lab to SLBMC and the communication of category 1 results to parents/guardians by SLBMC met achievable standards, the timeliness of the reporting of category 1 results by the lab in Guadeloupe to SLBMC and the communication of negative and or category 2 results by SLBMC to parents/guardians were outcomes that required improvement. Processes and competing priorities that might have existed due to the COVID-19 pandemic warrant review. The importance of educational programs was evident, as 97% of fathers were unaware of their sickle cell status. It must be noted that SCD NBS does not end at the receipt and dissemination of results but rather should ensure that babies are followed through to their first follow-up by a specialist/local centre to include the timely institution of preventative care. Through the after of QI teams and programs, a review of SCD morbidity and mortality 5 years post the initiation of the SCD NBS screening program would be beneficial. Robust educational campaigns have the scope to enable vast improvements. The sustainability of the program is heavily dependent on the availability of grants; thus, it may be prudent to explore the use of POC testing.

## Figures and Tables

**Figure 1 IJNS-09-00014-f001:**
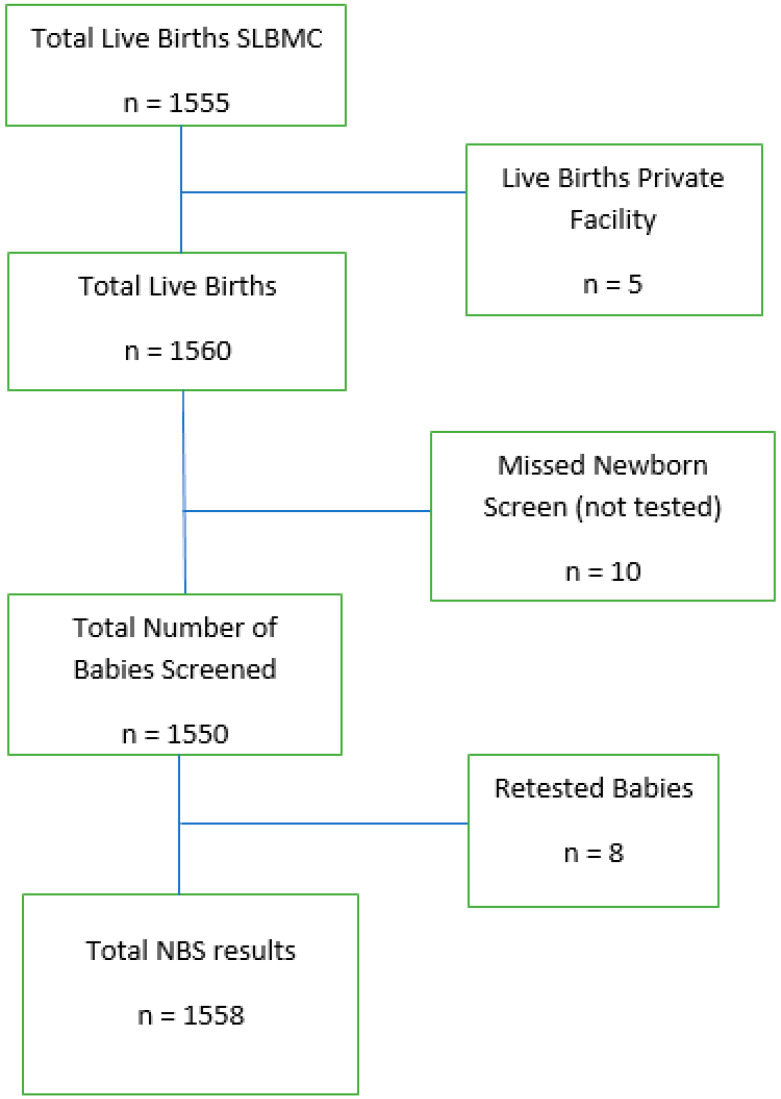
Flow chart of number of subjects included in audit.

**Figure 2 IJNS-09-00014-f002:**
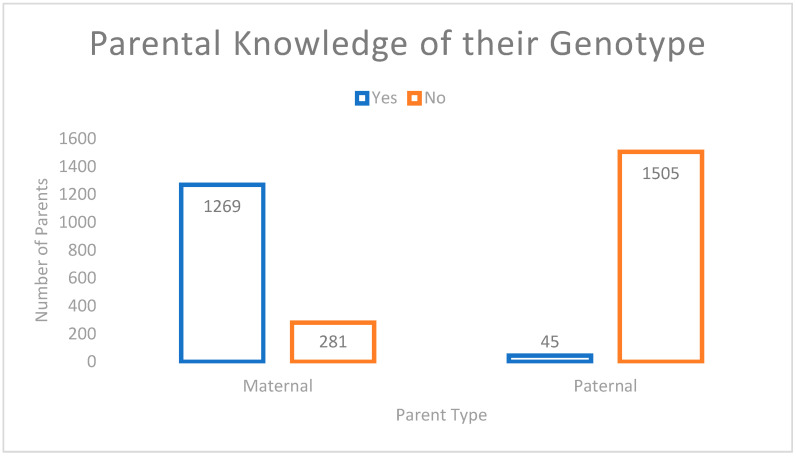
Parental knowledge of their Hb Electrophoresis status at time of newborn screen.

**Table 1 IJNS-09-00014-t001:** Frequency of results of newborn screen.

Phenotype	Frequency (*n*)	Percentage (%)
FA	1313	84.3
FAS	150	9.6
FAC	71	4.6
Inconclusive/Contaminated	11	0.7
FS	4	0.3
FAX *	4	0.3
FSC	3	0.2
FC	2	0.1
	1558	100

* FAX—abnormal Hb different from HbS and HbC.

**Table 2 IJNS-09-00014-t002:** Standards of reporting of results by lab in Guadeloupe to SLBMC, and communication of results by SLBMC to parents.

Standard Number	Standard	Acceptable Standard	Achievable Standard	Current Standard (Compliance)
-	Reporting of Category 1 Result to SLBMC by Guadeloupe Laboratory by 20 days	90%	95%	80% Maximum: 27 days Minimum: 14 days Mean 19 days
NP3	Communication of Category 1 results to parent/guardian by 4 weeks	90%	95%	100%
-	Reporting of Category 2 results, negative and other results to SLBMC by Guadeloupe Laboratory by one month	90%	95%	100%
-	Communication of category 2, negative screening and other results to parent/guardian by one month	90%	95%	54%

**Table 3 IJNS-09-00014-t003:** Breakdown of parental genotype based on maternal antenatal record and paternal self-reporting.

Genotype	Maternal		Paternal	
	Frequency (*n*)	Percentage (%)	Frequency (*n*)	Percentage (%)
AA	1088	70	31	2
Unknown	281	18	1505	97
AS	133	9	12	1
AC	45	3	2	-
SS	2	-	0	0
SC	1	-	0	0
TOTAL	1550	100	1550	100

**Table 4 IJNS-09-00014-t004:** Paternal knowledge of sickle status based on mother being positive/a carrier.

		Maternal SCD/CARRIER State			Test—Chi Squared (Uncorrected)
Paternal knowledge of SCD status		YES	NO	TOTALS	21.3
	YES	15	30	45	
	NO	165	1340	1505	
	TOTALS	180	1370	1550	*p* value (1 tail) 0.000001967

## Data Availability

The data presented in this study are available on request from the corresponding author. The data are not publicly available due to privacy restrictions.

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
