# Peer review of "Sickle Cell Disease Newborn Screening—An Audit of a Twin Island State Pilot Program"

_2409-515X, 2023, doi:10.3390/ijns9010014_

Round 1

Author Response

Page 3 of 12 line 48 has been corrected as follows: ... that are in the "Recommended Uniform Screening Panel"...

Words in parenthesis were capitalized as they represent a formal noun.

Reviewer 2 Report

This is an important manuscript describing the audit of a newborn screening program, in place in Antigua and Barbuda. The authors present the data clearly.

I have a few questions:

-IS the program supported publicy or do parents have to pay?

-Were there technical issues in managing the samples? Or in performing the analyses?

-How do the authors plan to address the lack of knowledge by male partners regarding their status? Are public awareness campaigns planned?

Author Response

  1. The program is supported publicly, parents do not have to pay
  2. There were no significant technical issues in managing the samples. Once taken they were allowed to air dry for at least 4 hours, allowed to air dry, and then refrigerated. The only challenge reported by lab was when contamination of a sample was suspected, in which case they would suggest that a repeat sample be sent. There are three main situations in which a sample result is deemed to be inconclusive a. damaged sample, b. low haemoglobin level or low blood collection, c. haemoglobin A and F levels inconsistent with Gestational Age
  3. Public awareness campaigns are planned.  Educational sessions for health care providers to include Urologists are also planned where males may be captured during their check-up. Pre-marital testing will also be an area of focus.